# Reduced Sulfur Diet Reshapes the Microbiome and Metabolome in Mild–Moderate Ulcerative Colitis

**DOI:** 10.3390/ijms26104596

**Published:** 2025-05-11

**Authors:** Jiayu Ye, Maitreyi Raman, Lorian M. Taylor, Munazza Yousuf, Remo Panaccione, Christian Turbide, Sidhartha R. Sinha, Natasha Haskey

**Affiliations:** 1Division of Gastroenterology and Hepatology, Stanford Medicine, Stanford University, 300 Pasteur Dr., Palo Alto, CA 94305, USA; jyye@stanford.edu (J.Y.); sidsinha@stanford.edu (S.R.S.); 2Department of Medicine, Cumming School of Medicine, University of Calgary, 2500 University Drive NW, Calgary, AB T2N 1N4, Canada; mkothand@ucalgary.ca (M.R.); lorian.taylor@lyfemd.com (L.M.T.); munazza.yousuf1@ucalgary.ca (M.Y.); rpanacci@ucalgary.ca (R.P.); christian.turbide@mac.com (C.T.); 3Department of Biology, Irving K Barber Faculty of Science, University of British Columbia-Okanagan, 3187 University Way, Kelowna, BC V1V 1V7, Canada

**Keywords:** sulfur, sulfur metabolites, sulfites, reduced sulfur diet

## Abstract

This pilot study investigated the effects of a reduced sulfur (RS) diet on the gut microbiome composition and fecal metabolome in individuals with remitted or active ulcerative colitis (UC). Thirteen participants maintained their habitual diet (control), while nine followed an RS diet for eight weeks (Wk8). Stool and plasma samples were collected at the baseline and Wk8. The sulfur intake decreased in the RS group (−28 g/1000 kcal) versus the control group (−1.7 g/1000 kcal; *p* < 0.001). The RS group exhibited a significant decrease in lipopolysaccharide-binding protein (−5280 ng/mL), while these levels increased in the control group (620 ng/mL; *p* < 0.05). The microbiome analysis showed an increased alpha diversity at Wk8 (*p* < 0.01), suggesting a microbial shift with a RS intake. The metabolic alterations indicated enhanced nitrogen disposal (increased uric acid, methyluric acid, N-acetyl-L-glutamate) and a higher energy demand (elevated ubiquinol and glucose-pyruvate). The RS diet increased beneficial microbes *Collinsella stercoris*, *Asaccharobacter celatus*, and *Alistipes finegoldii*, while decreasing pathobionts *Eggerthella lenta* and *Romboutsia ilealis*. Methyluric acid correlated positively with *C. stercoris* (β = 0.70) and negatively with *E. lenta* (β = −0.77) suggesting these microbes utilized this metabolite and influenced the microbiome composition. In conclusion, a RS diet promoted microbial diversity, metabolic adaptations, and reduced inflammation, highlighting its potential as a novel strategy for UC management.

## 1. Introduction

Sulfur is one of the most abundant minerals found in the human body and it is integrated into virtually every aspect of biochemical function. It is essential for the maintenance of cellular structure, cellular signaling within the organism, detoxification of free radicals, and assisting with energy production [1,2,3]. Diet is the primary source of sulfur, with key contributors including protein-rich foods that include sulfur-containing amino acids, like methionine and cysteine, as well as sulfites used as preservatives in processed foods and beverages [4]. No established daily requirements for sulfur have been determined.

Sulfur metabolism and the role of sulfur-containing foods in health are complex. While sulfur is essential for life, excessive levels can pose potential health risks. Among sulfur metabolites, hydrogen sulfide (H_2_S) has garnered significant interest for its impact on gastrointestinal health and the microbiome. H_2_S is primarily produced by intestinal microbiota, particularly sulfate-reducing bacteria (SRB) [2,3,5]. Under normal physiological conditions, intestinal epithelial cells efficiently metabolize H_2_S, where it contributes to mucosal defense, regulates hepatic and mucosal blood flow and supports overall gut homeostasis [2,6]. However, when the intestinal barrier is compromised, elevated H_2_S levels can become toxic and have been implicated in inflammatory bowel disease (IBD) [7]. Excess H_2_S adversely affects the intestinal mucosa, which inhibits the growth of colonocytes and disrupts colonic cell metabolism by impairing butyrate oxidation, which is essential for epithelial barrier integrity, ultimately contributing to tissue injury [2,8,9].

There is considerable evidence that links SRB and H_2_S to the pathogenesis of ulcerative colitis (UC). Elevated levels of SRB and H_2_S have been consistently observed in the feces of UC patients [10,11,12,13], with concentrations significantly higher during active disease compared with periods of remission [11,14]. The correlations with disease activity further support SRB’s role in UC pathogenesis. The therapeutic efficacy of 5-aminosalicylic acid (5-ASA) and its prodrug, sulfasalazine, further supports this connection, as both inhibit SRB and H_2_S production in the colon. 5-ASA reduces inflammation by blocking cyclooxygenase and lipoxygenase pathways, while sulfasalazine, upon bacterial cleavage, releases 5-ASA and sulfapyridine [11]. By suppressing SRB growth and limiting H_2_S toxicity, these agents mitigate mucosal damage in UC.

Dietary factors may further influence the growth of SRB and H_2_S production. In vitro studies indicate that SRB proliferation can be stimulated by sulfur-rich amino acids commonly found in protein-rich diets [4,15]. A Western diet—characterized by high protein and low fermentable fiber—can promote the production of harmful compounds, like H_2_S, while also being associated with a lower production of beneficial metabolites, such as butyrate [9]. Dietary sources of sulfur are diverse and include both naturally occurring foods and food additives. Protein-rich foods, such as meat, poultry, fish, eggs (particularly yolks), and dairy products, are significant contributors due to their content of sulfur-containing amino acids, like methionine and cysteine [4]. Plant-based sources include cruciferous vegetables (e.g., broccoli, cauliflower, cabbage, Brussels sprouts, kale, arugula), which are rich in glucosinolates, and allium vegetables (e.g., garlic, onions, leeks, shallots, chives), which contain organosulfur compounds, like allicin [4]. Legumes and soy products also provide moderate amounts of sulfur. Additionally, processed foods and beverages—including certain breads; soya flour; dried fruits; brassicas; processed meats; and alcoholic drinks, like beer, cider, and wine—are notable sources, particularly due to added sulfites used as preservatives [4]. Common food additives, such as sulfur dioxide, carrageenan, agar, xanthan gum, and ammonium compounds, are frequently found in ultra-processed foods and beverages, which have been implicated in the development and adverse disease course of inflammatory bowel disease (IBD) [16].

Although dietary interventions can modulate the gut microbiome, the relationship between sulfur-containing foods and UC remains insufficiently explored. Few studies have directly assessed the therapeutic potential of modifying dietary protein or sulfur intake to induce a clinical response or prevent relapse in UC, and while initial findings are promising, small sample sizes limit their conclusions [17,18,19,20,21]. To our knowledge, no studies have investigated the impact of a reduced sulfur diet on the gut microbiota and metabolome. This study aimed to determine whether incorporating a reduced sulfur diet alongside conventional medical management in patients with mild-to-moderate UC leads to changes in disease activity, as well as alterations in the gut microbiota and metabolome, compared with those maintaining their usual dietary intake with conventional medical management. 

## 2. Results

### 2.1. Participant Information

The baseline characteristics of the study participants are summarized in Table 1 (Appendix A). A statistically significant difference was observed in the sex distribution, with a greater proportion of female participants in one group (*p* = 0.0001). The median age of the participants in the RS group was 41 years [IQR 32–46], where most participants had a partial Mayo score (pMs) indicating mild disease activity or remission (89% with a pMS of 0–4) and were female (78%). The median age of the control group was 47 years [IQR 29–50], where 62% had a pMS of 0–4 and were female (38%). The baseline age, body mass index (BMI), fecal calprotectin, and medications were similar between the groups. The control group exhibited higher disease activity compared with the RS group. Specifically, the control group had more participants with moderate-to-severe pMS (38%) versus 11% in the RS group.

### 2.2. Diet Characteristics of the Study Participants

Next, we examined the participants’ diets to identify the potential differences in the nutrient intake between the RS and control groups. No significant differences were observed within or between groups in the adjusted macronutrient intake (calories, protein, fat carbohydrates, sugar, and fiber), micronutrients (calcium, iron, vitamin B12, choline, or zinc), the Mediterranean Diet Adherence Score [22], and the Total Healthy Eating Index-2020 (HEI) [23] or HEI moderation components (*p* > 0.05) (Figure 1A,B, Appendix A). The change in sulfur intake (adjusted/1000 kcal) from the baseline to week 8 in the reduced sulfur (RS) group was −29 g/1000 kcal [IQR: −49 to −21], whereas the change in the sulfur intake (adjusted/1000 kcal) in the control group was −1.7 g/1000 kcal [IQR: −24 to 36] (*p* = 0.003) (Figure 1C). At the baseline, the total daily intake of sulfur in the RS group was a median of 411 mg/day (IQR: 292–539) and decreased to a median of 172 mg/day (IQR: 140–273) (*p* < 0.01) (Figure 1D). Figure 1E illustrates the percentage contribution of various food sources to total sulfur intake at the baseline and after eight weeks. At the baseline, grain products, water, and condiments were the primary sources of sulfur, contributing 22%, 13%, and 16% of the total intake, respectively. By week 8, the contribution from grain products significantly decreased to 10% (*p* = 0.005), while water increased significantly to 16% of the total intake (*p* = 0.007). Condiments increased, but this was not statistically significant. Notably, the intake of sulfur from mixed dishes, nuts and seeds, vegetables, fruit, alcohol, and milk products and substitutes remained stable (*p* >0.05). Processed meats, which contributed 2% at the baseline, dropped to 0% at week 8. Meanwhile, the protein intake (excluding processed meats) increased from 2% to 5%, suggesting a shift toward different protein sources.

### 2.3. Reduced Dietary Sulfur Intake Improved Disease Activity and Influenced Short-Chain Fatty Acid Production

We analyzed the clinical biomarkers to identify the potential differences between the groups. A fecal calprotectin response, defined as a ≥50% reduction from the baseline to week 8, was observed in 43% (n = 3/7) of the RS group compared with 25% (n = 3/12) of the control group (*p* = 0.01) (Figure 2A). Additionally, the novel biomarker lipopolysaccharide-binding protein (LBP)—which has been associated with disease activity markers and dietary factors in UC [3]—was significantly reduced in the RS group compared with the controls (−5280 [IQR: −12405 to 2390] vs. 620 [−2260 to 10225], *p* = 0.04) (Figure 2B). The partial Mayo score decreased from 2 to 0 [IQR: 0–1.5] in the RS group (*p* = 0.06) and from 4 to 1 [IQR: 0–2.5] in the control group (*p* = 0.03) (Figure 2C). In contrast, the zonulin levels showed no significant differences within or between the groups (Appendix A).

An analysis of the short-chain fatty acid levels revealed a trend toward increased concentrations of butyric acid (fold change) in the RS group compared with the control group (*p* = 0.06) (Figure 2D). The valeric acid levels (fold change) were significantly elevated in the RS group compared with the control group (*p* = 0.03) (Figure 2E). The total short-chain fatty acids, acetic acid, iso-butyric acid, and propionic acid levels did not differ between the groups (Appendix A).

### 2.4. A Reduction in the Dietary-Sulfur-Induced Metabolic Shifts

An untargeted metabolomic analysis of fecal samples by Partial Least Squares Discriminant Analysis (PL-SDA) revealed a clear separation between the baseline and week 8 timepoints in the RS group (Figure 3A). In contrast, no comparable separation was observed between the baseline and week 8 in the control group (Figure 3B). Moreover, the top markers that contributed to the separation showed only a weak correlation with the timepoints (|Spearman correlation| < 0.4, Appendix A). Further statistical analysis identified 22 significantly altered metabolites in the RS group, with 20 metabolites upregulated and two downregulated at week 8 (Appendix A). In the control group, nine metabolites showed significant changes, where four increased and five decreased at week 8 compared with the baseline (Appendix A). However, the changes in the control group were less pronounced, as indicated by weaker fold changes and a low correlation with time (Spearman < 0.4). In contrast, the RS group exhibited stronger correlations (Spearman > 0.5), suggesting more robust metabolic shifts (Appendix A). In the RS group, uric acid, methyluric acid, and N-acetyl-L-glutamic acid, which are all involved in the urea cycle, were elevated at week 8, suggesting enhanced nitrogen disposal. Additionally, 4-oxoproline, an intermediate of glutathione metabolism, was increased, indicating a potential adaptation to oxidative stress. The increased levels of glucose-pyruvate and QH_2_ (ubiquinol) suggest a heightened energy demand, likely reflecting metabolic adaptations to the dietary intervention (Figure 3C). Additionally, we observed an upregulation of microbiome-associated metabolites, including pyrocatechuic acid, indoleacetyl-glutamine, and N-eicosapentaenoyl phenylalanine, indicating that the reduced sulfur intake significantly modulated the gut microbial community (Figure 3C). The rise in guanosine and its related metabolites may reflect enhanced microbial proliferation and activity, further underscoring the influence of sulfur metabolism on microbiome dynamics.

### 2.5. Reduced Dietary Sulfur Induced Microbiome Shifts and Enhanced Diversity

Given the observed alterations in the metabolites that are primarily produced or metabolized by the gut microbiota, we next assessed the changes in microbiome composition. Our analysis revealed a significant increase in the alpha diversity, including the evenness and the Shannon index, at week 8 in the RS group compared with the control group (Figure 4A). Due to the small sample size in the RS group, we selected the top altered taxa with *p* < 0.1, ensuring that at least half of the samples (N ≥ 8) had nonzero values. This approach identified nine differentially abundant taxa (Figure 4B), with three taxa showing increased abundance and six decreased at week 8 in the RS group.

To further characterize the biologically relevant microbial alterations, we applied a fold change (FC) analysis, which excluded taxa where the FC values were not computed due to division by zero or there were missing values for abundance data. Using a cutoff of FC > 1.5 or <1/1.5, four taxa remained, all of which were present in the list of statistically significant changes (Appendix A). Notably, all individuals consistently exhibited a decrease in *Eggerthella lenta* (*p* = 0.01). Although the specific microbial taxa that responded to the RS intervention varied between the individuals, the general trend indicated an overall increase in the SCFA-producing taxon, including *Bacteroides ovatus*, *Parabacteroides distasonis*, *Streptococcus salivarius*, *Blautia wexlerae*, *Agathobaculum butyriciproducens*, *Faecalibacterium prausnitzii*, and *Flavonifractor plautii*. Additionally, *Ruthenibacterium lactatiformans*, a lactate producer, increased at week 8 in one patient (Table 2). This suggests that while the microbiome responses were personalized, the metabolic adaptation to sulfur management could contribute to a beneficial gut environment.

### 2.6. Eggerthella Lenta and Collinsella Stercoris Played Opposing Roles in Gut Metabolism and Inflammation

To examine the potential microbiome-metabolite associations, we applied a Spearman correlation analysis of the top microbiome taxa (Figure 4B) and top metabolites (Figure 3C), with a cutoff of |ρ| > 0.6 (Appendix A, Figure 4C). *Eggerthella lenta* and *Collinsella stercoris* exhibited opposing correlations with several key metabolites. Both taxa were significantly associated with methyluric acid (*p* < 0.05), noradrenochrome (*p* < 0.05), and pyridoxolactone (*p* < 0.05), with *Eggerthella lenta* showing negative correlations and *Collinsella stercoris* showing positive correlations with these metabolites (Appendix A, Figure 4C). While *Eggerthella lenta* has been linked to gut inflammation, *Collinsella stercoris* may contribute to gut homeostasis, though its precise role in IBD is not clear [24]. *Eggerthella lenta* has also been negatively correlated with glucose-pyruvate, indicating a possible role in host energy metabolism, as well as guanosine and hydroxydeoxyguanosine, suggesting its involvement in microbial growth patterns. Overall, these findings suggest that reduced dietary sulfur could drive microbial adaptations, influencing metabolic pathways relevant to gut health and inflammation.

### 2.7. Dietary Choices Influenced Sulfur-Metabolizing Bacteria

A previous study has identified a list of 43 sulfur-metabolizing bacteria (SRB) [25]. Using food recalls for each participant in the RS group, the change in SRB from the baseline to Wk8 was correlated with the change in sulfur consumption stratified by food group from the baseline to Wk8 (Figure 5A). A high dietary intake of meat (chicken, beef, pork), eggs, and processed meats positively correlated with *Bilophila wadsworthia* and *Parabacteroides distasonis*. In contrast, the vegetables (including cruciferous vegetables), nuts and seeds, and legumes consumption showed negative correlations with sulfur-metabolizing bacteria, suggesting a potential protective effect. Water intake was generally negatively correlated with most bacteria, while alcohol was positively correlated with most sulfur-metabolizing bacteria.

## 3. Discussion

Many unknowns remain regarding sulfur requirements and their dietary provision in UC, where limited studies have explored how diet composition influences the structure and functional capacity of the microbial community. This study provides an initial demonstration that a reduced sulfur (RS) diet induces changes in the gut microbiome and metabolome, with a potential signal toward clinical improvements.

Dietary patterns play a crucial role in shaping the gut microbial composition, particularly sulfur-metabolizing bacteria linked to inflammation and conditions like IBD. This study is unique in that we examined the overall diet composition using the MEDAS [22] and HEI-2015 [23], in addition to sulfur-containing foods. Previous research on the H_2_S toxin hypothesis primarily focused on the sulfur-containing food intake, overlooking broader dietary patterns [9]. We found no significant differences in the diet composition scores between the groups; rather, the reduced sulfur intervention appeared to drive changes in the microbiota and metabolome.

A key finding is the strong positive correlation between the high consumption of meat and eggs with sulfur-metabolizing bacteria. This suggests that diets rich in animal-based proteins may promote the growth of bacteria that contribute to gut dysbiosis and inflammation. This aligns with previous research linking a high-sulfur amino acid intake (e.g., taurine, methionine) from animal sources to an increase in H_2_S-producing bacteria, which may exacerbate conditions like IBD [26].

Beyond animal products, sulfur is also abundant in plant-based foods, including cruciferous vegetables (e.g., broccoli, cauliflower), alliums (e.g., garlic, onions), and legumes. Additionally, sulfites—commonly used as preservatives in dried fruits, wine, and processed foods—contribute to dietary sulfur intake and may influence the gut microbial composition. Interestingly, we observed a negative correlation between vegetables (including cruciferous vegetables), nuts, seeds, legumes, and H_2_S-producing bacteria, suggesting that plant-based foods may help maintain the microbial balance and reduce pro-inflammatory bacterial populations. The fiber and polyphenols in these foods likely promote the growth of beneficial gut bacteria while suppressing sulfur-metabolizing species, thereby supporting intestinal homeostasis. These findings underscore the importance of dietary choices in shaping the gut microbial composition and inflammation risk, highlighting the need for a more nuanced approach to dietary sulfur intake beyond just meat consumption.

A notable reduction in *Eggerthella lenta*, which is known as a disease-causing microbe and associated with gut dysbiosis and inflammation, was observed at week 8 in the RS group, indicating a beneficial microbial shift [26]. In contrast, *Collinsella stercoris*, which is known for modulating the gut pH and promoting microbial stability, increased [24]. Despite individual variability, there was a broader expansion of SCFA-producing bacteria over time, including *Bacteroides ovatus*, *Parabacteroides distasonis*, *Streptococcus salivarius*, *Blautia wexlerae*, *Agathobaculum butyriciproducens*, *Faecalibacterium prausnitzii*, *Adlercreutzia equolifaciens*, and *Flavonifractor plautii*. Butyrate-producing microbial communities are essential for maintaining a healthy gut environment, and strong evidence supports short-chain fatty acid (SCFA) production as a key marker of gut ecosystem balance [27,28]. The expansion of SCFA-producing bacteria, particularly *F. prausnitzii*, *Adlercreutzia equolifaciens*, and *B. wexlerae*, increased the SCFA availability—especially butyrate—which may have driven this metabolic shift. SCFAs fuel ATP production through oxidative phosphorylation, and the rise in QH_2_, a critical component of the electron transport chain, which further supports increased mitochondrial activity.

Among the significantly altered metabolites between the baseline and week 8, indoleacetyl glutamine (IAG) and N-eicosapentaenoyl phenylalanine (NEP) were the two microbial-derived metabolites that increased following the sulfur restriction. IAG, a derivative of indole-3-acetic acid (IAA) and a microbial metabolite of tryptophan metabolism, belongs to the indole derivative family, which is known for its anti-inflammatory effects via activation of the Aryl Hydrocarbon Receptor (AhR) [29]. AhR, which is highly expressed at mucosal surfaces, plays a crucial role in enhancing the intestinal barrier function and promoting regulatory immune responses [30]. Notably, Ahr expression is downregulated in the intestinal tissue of patients with IBD [30,31], and IAA is selectively downregulated in the serum of patients with active UC compared with healthy controls [32]. The observed increase in IAG suggests a microbiome-driven metabolic shift toward anti-inflammatory pathways, potentially supporting mucosal homeostasis. NEP belongs to the N-acylamide family—a class of bioactive lipids formed by the conjugation of fatty acids and amino acids. These molecules are involved in immune regulation, metabolism, and gut–brain signaling, and some are partially produced or modified by gut microbes [33]. The increase in NEP after sulfur restriction suggests a potential shift in microbial lipid metabolism, which may influence gut homeostasis and inflammatory resolution. Together, the upregulation of IAG and NEP support the idea that microbiome-mediated metabolic adaptations may play a role in the beneficial effects of sulfur restriction in UC patients.

An altered purine metabolism, influenced by the diet and microbiota, has been implicated in the development of inflammatory bowel disease (IBD), with evidence suggesting that its regulation is crucial for disease management [34]. Recent research highlights the role of microbiota-derived purines as key substrates salvaged by the colonic epithelium for nucleotide synthesis and energy balance, which are both essential for epithelial homeostasis and wound healing [35]. We observed an increase in stool uric acid in the RS group at week 8. As the final breakdown product of purine metabolism, uric acid is primarily generated in the liver and excreted via the kidneys, with the intestine serving as an auxiliary elimination route [36]. While the gut does not typically produce uric acid, this finding suggests enhanced intestinal clearance, possibly mediated by ABCG2 transporter activity—an alternative pathway that compensates when renal excretion is insufficient [37]. This shift may improve the systemic nitrogen balance by diverting excess nitrogenous waste toward intestinal excretion rather than local accumulation, potentially mitigating the inflammatory burden. Clinically, the thiopurine immunosuppressant azathioprine (AZA) is widely used to induce and maintain remission in moderate-to-severe IBD by inhibiting de novo purine synthesis, thereby limiting the immune cell proliferation [38,39]. In contrast to this synthetic blockade, the observed rise in stool uric acid may reflect increased purine catabolism, promoting immune resolution by actively clearing excess purines. Thus, elevated stool uric acid could serve as a metabolic marker of purine turnover and immune remodeling, paralleling AZA’s therapeutic effects through an endogenous, catabolic mechanism.

The observed increase in N-acetyl-L-glutamic acid (NAG) at week 8 in the RS group further supports an upregulation of the urea cycle, facilitating the conversion of ammonia into urea for excretion and preventing nitrogen overload [40]. Ammonia accumulation is known to contribute to increased intestinal permeability, oxidative stress, and gut inflammation [41]. These metabolic shifts suggest that a reduced sulfur intake enhances nitrogen clearance, potentially lowering the inflammatory burden by improving the intestinal permeability, as indicated by lower levels of lipopolysaccharide-binding protein (LBP) in the RS group. LBP is an emerging marker for intestinal permeability [42]. Furthermore, the concurrent rise in guanosine and hydroxyguanosine, which are purine-derived metabolites, indicates active purine turnover without excessive breakdown or systemic accumulation, suggesting that nitrogen metabolism was optimized rather than dysregulated. Previous studies have demonstrated that limiting sulfur-containing amino acids reduces serum uric acid levels, further supporting the hypothesis that sulfur metabolism influences purine degradation and uric acid production [43]. Taken together, the increase in fecal uric acid following sulfur restriction suggests a shift in uric acid clearance from systemic circulation to intestinal excretion. Given the role of excess ammonia in oxidative stress and inflammation, these metabolic adaptations may contribute to an improved nitrogen metabolism and reduced systemic and intestinal inflammatory burdens in UC patients.

Guanosine is a guanine-based purine that functions as an extracellular signaling molecule with documented anti-inflammatory and antioxidative effects [44,45]. The gut microbiota actively metabolizes purines, including guanosine, through nucleotide recycling and salvage pathways, which are crucial for microbial growth and ecosystem stability [45]. In experimental models, exogenous guanosine supplementation has been shown to exert anti-inflammatory effects by modulating adenosine signaling and mitochondrial function [44]. In our study, stool guanosine levels significantly increased in the RS group at week 8, likely reflecting the enhanced microbial turnover and metabolic activity. Given that stool nucleotides primarily originate from the gut microbiota rather than host tissues, this rise suggests shifts in the microbial composition and inflammatory status. Notably, the increase in stool guanosine coincided with a greater microbiome alpha diversity, indicative of a more dynamic and metabolically active microbial community.

The current study had several limitations. As an exploratory investigation, our primary goal was to assess the initial clinical and microbiome responses to a reduced sulfur diet; however, future studies will aim to quantify the total sulfur intake and evaluate relevant biomarkers—such as elemental sulfur levels in biological samples—using advanced techniques, like ICP-MS, to better understand sulfur’s role in ulcerative colitis. One notable limitation is the absence of an objective measure of disease progression, such as endoscopy in all patients, which limits the ability to complement the clinical biomarkers used. Additionally, the relatively short duration of the low dietary sulfur intervention may not be sufficient to capture meaningful clinical responses, such as endoscopic or histological changes, which typically require longer periods to manifest in dietary studies. The inherent complexity of the microbiome also presents challenges, as individual variations in microbial composition may lead to heterogeneous responses to dietary modifications. Furthermore, the sulfur contents of foods vary widely, and reliance on food diaries or self-reported intake may not fully account for this variability, reducing the precision of the dietary assessment. Lastly, as this study was underpowered, future research should incorporate a larger sample size and a longer intervention period to determine whether microbiome alterations translate into sustained clinical benefits, such as reduced inflammation.

## 4. Materials and Methods

### 4.1. Study Design and Participants

This post hoc analysis was based on an open-label, 8-week randomized controlled trial that investigated the effects of reduced dietary sulfur (RS) on the gut microbiota, metabolome, and clinical disease in ulcerative colitis (UC). Details of the original trial’s study design and the CONSORT study flow diagram are available elsewhere [46].

This study included participants with both baseline and week 8 (Wk8) fecal samples, which allowed for the assessment of sulfur’s influence on the microbiome and metabolome (n = 22) [46]. The participants were further stratified into those who reduced their sulfur/sulfate intake (RS group) (n = 9) and those whose sulfur/sulfate intake remained unchanged (controls, n = 13). To account for variations in the caloric intake, we used the adjusted sulfur intake per 1,000 kcal.

### 4.2. Inclusion and Exclusion Criteria

This study was conducted in the outpatient IBD clinic at the University of Calgary from April 2016 to January 2021. The inclusion criteria for the intervention trial were adult patients (age > 18 years) with UC established by the usual endoscopic and histologic criteria. Those with a clinical disease flare, defined as a partial Mayo score (PMS) > 2, were managed by their gastroenterologist with any combination of 5-ASA (oral, topical, or both), an immunomodulator, oral corticosteroids, or the initiation of a new biologic as part of their usual care. The participants with UC in clinical remission were defined by PMS ≤ 2 and maintained on any combination of the therapies mentioned above. The exclusion criteria included major medical comorbidities (diabetes, active malignancy within the past five years, active infections, severe respiratory or cardiac disease, acute or chronic kidney disease), previous bowel surgery, or smoking. No adverse events were reported.

### 4.3. Reduced Sulfur Intervention

The intervention group received conventional medical management plus a reduced sulfur (RS) diet and diet counseling by an RD. Sulfide is generated in the large intestine in two ways: first, by the action of sulfate-reducing bacteria on inorganic sulfates from foods, including dried fruits, cruciferous vegetables, nuts, and fermented beverages, and second, by the fermentation of organic sulfur-containing amino acids found in red meat, seafood, eggs, milk, and cheese [5,47,48]. Drinking water can also contribute significant amounts of sulfate to the diet, where concentrations ranging from 10 mg/L to 1795 mg/L have been reported in the Canadian water supply [49].

The diet education was delivered by a registered dietitian, and each patient was provided with an individualized plan. An RS diet included the reduction of food and food additives (e.g., carrageenan) and beverages high in sulfate/sulfur [48,50], consuming drinking water low in sulfate (can be as high as 500 mg/L in Alberta so identifying the drinking water source was necessary) [49], and limiting sulfur-containing supplements (e.g., chondroitin sulfate). The RS diet eating plan, resources on reduced sulfur/sulfate eating, and an RD counseling session were designed and reviewed by experts in nutrition, dietary design, education resources, and dietary behavior change. The control group received one session with the RD on how to reduce sulfur in the diet at the end of 8 weeks. Both the RS and control groups received conventional management with visits to the clinic at 4 weeks to ensure completion of their 24 h food recalls and food frequency questionnaires and to discuss the disease course with their physician.

The food and beverage intake reported by the participants was assessed at the baseline and Wk 8 using two non-consecutive 24 h food recalls using the Automated Self-Administered 24 h (ASA-24^®^) Dietary Assessment Tool (Canadian version) [51]. Detailed food and beverage items, food groups, and nutrient intakes for macro- and micronutrients were downloaded from the ASA-24 researcher website. The data cleaning followed guidelines from the US National Cancer Institute [52]. Macro- and micronutrients were also evaluated in relation to the energy intake (per 1000 kilocalories), which were referred to as adjusted macro- and micronutrients. ASA-24 does not capture sulfur or sulphate intake; therefore, mean values were calculated for each participant using values described in the literature [47,48] (see Appendix A). The Mediterranean Diet Adherence Screener was used to calculate a Mediterranean diet score from the diet records [22].

### 4.4. Fecal Microbiome Analysis

The microbiome analysis was conducted at the International Microbiome Centre (IMC) at the University of Calgary (Calgary, Canada). The DNA isolation procedures and microbiome pipelines used are described elsewhere [46]. We extracted a subset of the data to include the participants with reduced sulfur (n = 9) versus those that did not change the sulfur intake (n = 13) and those that had both baseline and Wk8 fecal samples. We recalculated the alpha-diversity metrics (observed species, Shannon, Simpson) and β-diversity indices (Bray–Curtis, binary Jaccard) using the VEGAN package in R. The taxonomic differences between week 8 and the baseline within each group were analyzed using MaAsLin2 [53].

### 4.5. Fecal Metabolome Analysis

Fecal extractions and metabolic analysis according to the previously published methods [46,54,55,56,57]. The raw data were converted to the mzML format using MSConvert (v2.1, ProteoWizard) and processed via MetaboanalystR 4.0 [58]. A noise threshold of 1000 (peak intensity) was used when constructing the isolated chromatograms for each mass. The LOESS regression method was applied for the retention time (rt) correction, and missing values were imputed using local chromatographic signals. The CompoundDb package was used to obtain the HMDB v5.0 database (Human Metabolome Database), which was then loaded into R as an SQLite database [59]. The MetaboAnnotation package was subsequently utilized to annotate the MS1 data by matching the observed m/z values against the compounds in the HMDB database [58]. For the MS2 data annotation, the DDA method was selected by leveraging m/z and rt from the MS1 scans [58]. The method selected the top four most abundant ions per scan from the MS1 data for further MS2 analysis. The resulting MS2 spectra were then matched to metabolites in five embedded databases: HMDB, METLIN, LIPIDMAPS, KEGG, and MassBank, using a mass tolerance of 10 ppm.

### 4.6. Metabolomic Data Processing and Statistical Analyses

Statistical analyses were performed using R (v4.2.0). Non-normally distributed data were normalized using the area normalization method. Partial Least Squares Discriminant Analysis (PLS-DA) was used to assess the distributions. The metabolite selection was based on the combination of VIP (Variable Importance in Projection) scores, correlation analysis, fold changes (FCs), and *p*-values to ensure the selected metabolites were associated with different biological states. VIP scores (derived from the PLS-DA) highlight the most influential features contributing to group separation. Due to the non-normal distribution of the data, all correlation analyses in this study were conducted using the Spearman correlation coefficient. Clustering heatmaps were created using the z-score of the metabolite peak intensity, plotted with the Pheatmap package in R [60]. Given the non-normal distribution of the data, the Spearman correlation was used to assess the relationship between differential metabolites and clinical parameters, as calculated with the cor() function.

### 4.7. Short-Chain Fatty Acids

Short-chain fatty acids were quantified according to previous methods [46]. LC-MS/MS analysis and data analyses were performed according to Bihan et al. [61].

### 4.8. Statistical Methods

The Anderson–Darling test was used to assess the normality of the data. Continuous variables are presented as the mean and standard deviation (SD), or median and interquartile range (IQR), and categorical data are presented as the absolute value and percentage. The Wilcoxon matched-pairs signed-rank test (non-parametric) and paired *t*-test (parametric) were used for paired data, and the Mann–Whitney U test (non-parametric) or unpaired (*t*-test) was used for the comparison between the groups. Categorical variables were compared using Fisher’s exact test. The Spearman correlation was used to assess the association between the metabolites and the microbiome, as well as between the microbiome and sulfur contributions across the food categories. To better understand how the microbiome composition responded to the sulfur reduction from various food sources, delta values (Δ) were calculated for both the microbiome abundance and sulfur intake per food category. The Spearman correlation was then performed on these Δ values to identify the relationships between the changes in the microbiome composition and dietary sulfur intake. Taxa were selected for visualization if they exhibited at least one correlation coefficient (|ρ| ≥ 0.6) with any food category and had a corresponding *p*-value ≤ 0.1. The statistical package GraphPad Prism Version 10.4.1 (Graph Pad Software, San Diego, CA, USA) and R were used for the analyses and figures [62].

## 5. Conclusions

This study provided initial data that an RS diet induces notable shifts in the gut microbiome and metabolome, with potential implications for ulcerative colitis (UC) management. A key finding was the reduction of *Eggerthella lenta*, a microbe linked to gut dysbiosis and inflammation, alongside an expansion of SCFA-producing bacteria, including *Faecalibacterium prausnitzii* and *Blautia wexlerae*. These changes suggest an enhanced butyrate-driven metabolic environment that may promote gut homeostasis. Furthermore, the metabolic adaptations—including increased indoleacetyl glutamine (IAG) and N-eicosapentaenoyl phenylalanine (NEP)—point to microbiome-mediated anti-inflammatory mechanisms that could support intestinal barrier function.

The shifts in purine and nitrogen metabolism, such as increased fecal uric acid and N-acetyl-L-glutamic acid (NAG), suggest an improved nitrogen clearance and reduced systemic inflammatory burden. These findings highlight the potential of dietary sulfur modulation to influence microbial and metabolic pathways relevant to UC pathology and provide a foundation to inform future, adequately powered studies. Understanding the precise mechanisms linking dietary sulfur intake, microbial composition, and host metabolism will be crucial for developing targeted dietary strategies for UC management.

## Figures and Tables

**Figure 1 ijms-26-04596-f001:**
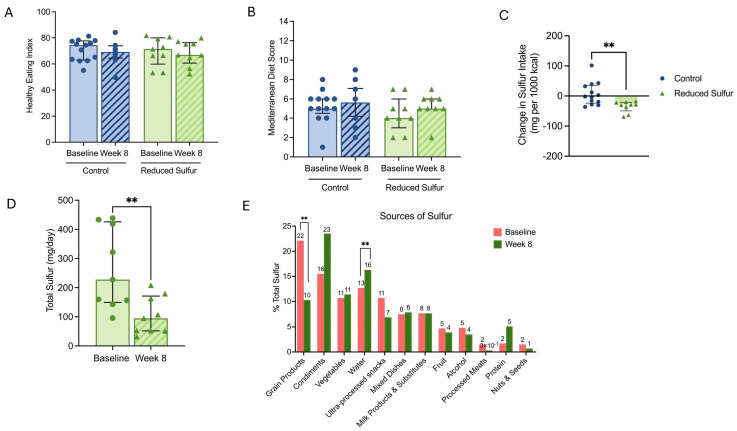
(**A**) Healthy eating index scores of the control (n = 13) and reduced sulfur (RS) (n = 9) groups. (**B**) Mediterranean diet adherence scores of the control and RS groups. (**C**) Change in the sulfur intake over time in the RS group compared with the control. (**D**) Total sulfur intake of the RS group at the baseline and week 8. (**E**) Comparison of the dietary sulfur intake sources at the baseline and after 8 weeks. The bar graph shows the percentage contribution of various food categories to the total sulfur intake per day at the baseline (salmon) and week 8 (green). The numerical values on the bars indicate the exact percentage contribution for each category. *p* < 0.05 was considered statistically significant. ** *p* < 0.01.

**Figure 2 ijms-26-04596-f002:**
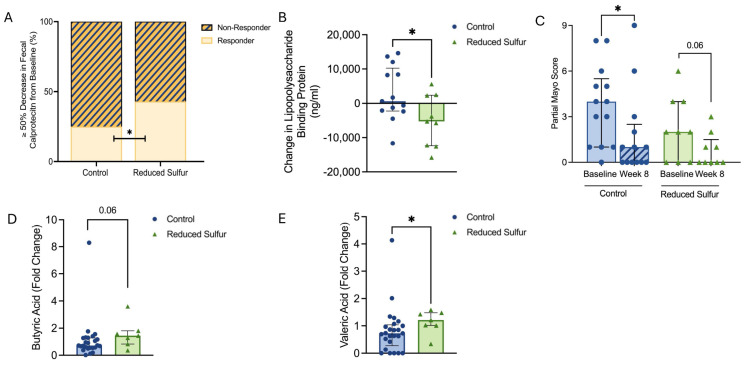
(**A**) Classification of the responders and non-responders based on a reduction of at least 50% in fecal calprotectin from the baseline. (**B**) Changes in the lipopolysaccharide-binding protein (LBP) levels over time between the control and reduced sulfur (RS) groups. (**C**) Changes in the partial Mayo scores from the baseline to week 8. (**D**) Fold change in the butyric acid levels between the control and RS groups, with increased levels in the RS group. (**E**) Fold change in the valeric acid levels between the control and RS groups, with increased levels in the RS group. *p* < 0.05 was considered statistically significant. * *p* < 0.05.

**Figure 3 ijms-26-04596-f003:**
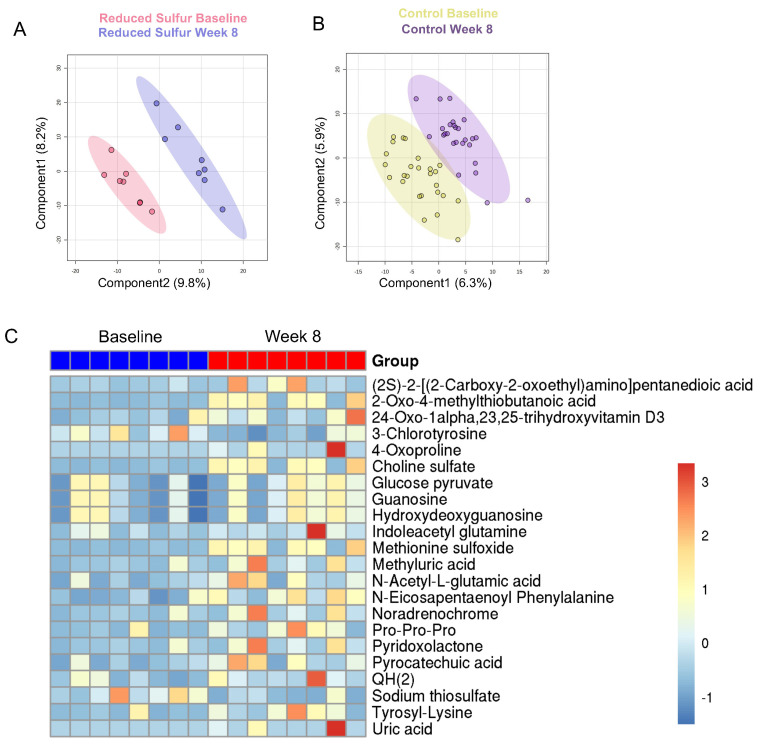
Untargeted metabolomic analysis of fecal samples using Partial Least Squares Discriminant Analysis (PLS-DA). (**A**) PLS-DA plot of the reduced sulfur (RS) group at the baseline (n = 8) and week 8 (n = 8). (**B**) PLS-DA plot of the control group at the baseline and week 8. (**C**) Heatmap comparison of the fecal metabolite profiles between the baseline and week 8 samples in the RS group. *p* < 0.05 was considered statistically significant.

**Figure 4 ijms-26-04596-f004:**
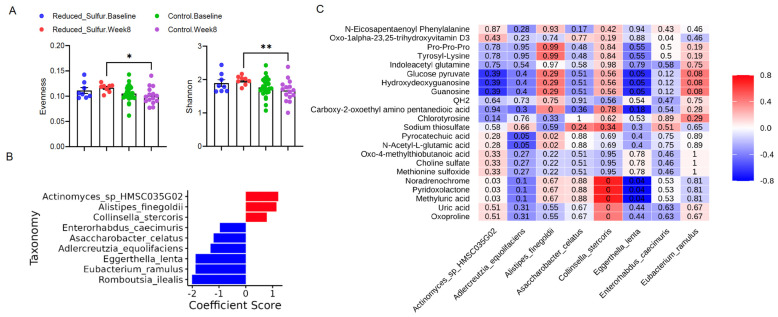
(**A**) Alpha diversity metrics (evenness and Shannon index) for the reduced sulfur (RS) group and control groups at week 8 (n = 8 per group). (**B**) Differentially abundant microbiota between the baseline and week 8 in the RS group (*p* ≤ 0.1); only taxa present in at least 50% of samples are shown. (**C**) Heatmap of correlations between the top metabolites and microbiota; the color gradient represents the strength and direction of correlations, with red indicating positive correlations and blue indicating negative correlations. Numbers represent *p*-values. *p* < 0.05 was considered statistically significant. * *p* < 0.05; ** *p* < 0.01.

**Figure 5 ijms-26-04596-f005:**
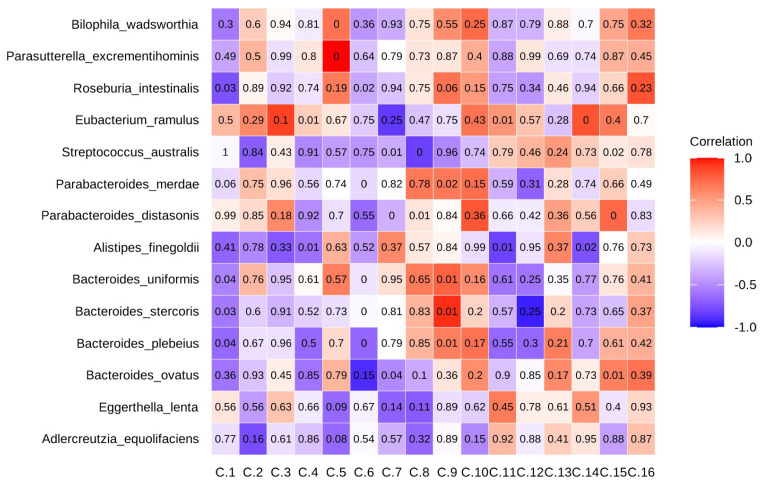
(A) This heat map visualizes the correlations between the dietary components (X-axis) and the sulfur-metabolizing bacteria (Y-axis). The color gradient represents the strength and direction of the correlations; red indicates a positive correlation (a stronger presence of bacteria with a higher intake of that food component) and blue indicates a negative correlation (a lower presence of bacteria with a higher intake of that food component). Numbers represent the correlation scores, and only correlations with *p* ≤ 0.1 are displayed. *p* < 0.05 was considered statistically significant.

**Table 1 ijms-26-04596-t001:** Baseline information about the study participants.

Participant Characteristics	Control (n = 13)	Reduced Sulfur Diet (n = 9)
Sex, female	5 (38%) *	7 (78%) *
Age (years)	47 (29–50)	41 (32–46)
Body mass index (kg/m^2^)	26 (25–31)	24 (22–28)
Partial Mayo score (n, %)		
Remission (<2)	4 (31%)	3 (33%)
Mild (2–4)	4 (31%)	5 (56%)
Moderate (5–7)	3 (23%)	1 (11%)
Severe (>7)	2 (15%)	0 (0%)
Fecal calprotectin (mcg/g)	593 (92–1789)	249 (44–1728)
Medications ^†^		
Steroids	5 (38%)	2 (22%)
5-ASA	9 (69%)	6 (67%)
Immunomodulator	1 (7%)	2 (22%)
Biologic	1 (7%)	2 (22%)

Data are given as median (IQR), n (%), or mean (SD). ^†^ Some percentages for medications exceeded 100%, as the participants may have been taking one or more medications. ** p* < 0.05.

**Table 2 ijms-26-04596-t002:** Fold change in microbiome taxa from baseline to week 8.

*Taxon*	*P52*	*P45*	*P44*	*P42*	*P30*	*P29*	*P28*
*Unknown*	1.0	1.0	1.2	0.9	1.0	0.9	0.9
*Adlercreutzia_equolifaciens*	1.5	0.5	0.4	0.4	0.1	0.6	0.4
*Asaccharobacter_celatus*	1.2	0.5	0.5	0.2	0.3	0.5	0.4
*Eggerthella_lenta*	0.5	0.4	0.3	0.4	0.0	0.3	0.6
*Enterorhabdus_caecimuris*	1.0	0.5	0.5	0.2	0.0	1.0	0.5
*Gordonibacter_pamelaeae*	1.4	0.5	0.7	0.1	3.7	0.6	0.6
*Bacteroides_ovatus*	2.5	2.6	0.3	0.4	0.0	3.2	30.9
*Parabacteroides_distasonis*	2.2	1.2	0.1	8.7	0.1	11.7	54.5
*Streptococcus_salivarius*	1.7	1.1	0.8	21.4	2.1	1.2	81.2
*Blautia_wexlerae*	1.8	1.9	2.2	0.1	0.5	0.1	1.5
*Agathobaculum_butyriciproducens*	2.2	0.3	7.4	4.3	4.3	0.3	0.2
*Faecalibacterium_prausnitzii*	1.0	2.3	1.8	11.5	0.8	4.4	0.4
*Flavonifractor_plautii*	3.7	0.0	0.1	4.5	0.2	0.0	1.5
*Ruthenibacterium_lactatiformans*	0.8	0.2	0.2	0.3	2.5	0.2	1.0
*Clostridium_spiroforme*	0.7	1.2	1.4	0.0	2.0	0.9	0.0

P52 refers to participant 52, P45 to participant 45, etc. The fold change (FC) values are presented as week 8/baseline. A fold change cutoff of 1.5 or 1/1.5 was applied, where blue indicates a decrease (FC < 1/1.5) at week 8 and red indicates an increase (FC > 1.5) at week 8. *p* < 0.05 was considered statistically significant.

## Data Availability

The raw data are openly available at Figshare: https://figshare.com/projects/Reduced_Sulfur_Diet_Reshapes_the_Microbiome_and_Metabolome_in_Mild-Moderate_Ulcerative_Colitis/243332 (accessed on 9 May 2025). The raw sequencing and metabolomic data will be provided upon request.

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
