# Peer review of "Reduced Sulfur Diet Reshapes the Microbiome and Metabolome in Mild–Moderate Ulcerative Colitis"

_ijms, 2025, doi:10.3390/ijms26104596_

Round 1

Reviewer 1 Report

Comments and Suggestions for Authors

1. The authors have investigated ulcerative colitis clinical responses from the perspective of the gut microbiome and metabolites by intervening in sulfur intake. The insights gained from this study are undoubtedly valuable; however, there is concern that some ambiguity and inconsistency in how the results are presented could diminish the study’s overall impact.

2. There appears to be an inconsistency in the figure legends: some primarily describe the figure’s format and statistical methods, while others delve into the clinical or medical interpretation of the data. For instance, Figure 5 tends to focus on explaining the figure format, whereas Figure 2 seems to emphasize interpretive aspects. Both are important, but balancing them is crucial. I personally recommend emphasizing the former (i.e., explaining the figure’s structure, color coding, statistical methodology) in the legends, while leaving detailed medical interpretation to the Results section. This approach would reduce potential confusion for readers.

3. Please exercise caution when indicating statistical significance in tables and figures. For example, in Table S1, the meaning of each P value is not entirely clear. Providing a concise explanation of the statistical tests performed and precisely what the P values represent would help readers interpret the results accurately.

4. Please ensure that every figure and table you have created is explicitly explained in the Results section. For instance, Figure 4C does not appear to be sufficiently addressed in the main text. Clarifying its content in the Results would greatly improve readability and coherence.

5. According to the text, there should be nine samples in the Reduced Sulfur group. However, in Figure 3A, there are only seven “Baseline” data points and eight “Week 8” data points plotted. Likewise, Figure 3C shows eight “Baseline” and seven “Week 8” points. If any samples were removed or excluded during the analysis, please clarify why. This transparency is essential for readers to understand the data fully.

6. In lines 164–165, the manuscript states that “no such separation was observed between baseline and week 8 in the control group” with reference to Figure 3B. However, based on the figure, the purple and yellow regions only overlap slightly, suggesting there may be a moderate degree of separation, at least for most samples. If the intent is to compare dietary intervention effects between the Reduced Sulfur group and the Control group, it might be helpful to use some form of discriminant analysis or quantitative index to illustrate this separation more objectively.

7. In Figure 3C, it is not immediately clear what the colors in the heatmap represent. Please consider adding a color scale or chart to clarify which statistical or biological quantity is being depicted by each color.

8. There are various clustering algorithms available. For Figure 3C, please clarify in the Methods section which algorithm (e.g., hierarchical clustering with a specific linkage method) was used. This additional detail will help readers fully understand how the clusters were generated.

9. The x-axis in Figure 4B is not clearly explained. Please specify what statistical or biological measure it represents, such as fold change, effect size, or another metric.

10. In Figure 4C, the numeric values inside each cell are described as P values, while the color scale indicates the strength of positive (red) or negative (blue) correlations. However, there appear to be cells where the color suggests a strong correlation yet the P value is very high (e.g., 0.99), and others that look nearly neutral in color but report a significant P value (e.g., 0.03). Please clarify why these apparent discrepancies occur so readers can better interpret the data.

11. In Figure 4C, there is a dendrogram for the metabolites, whereas in Figure 5 there is no dendrogram. Also, the numerical values in each cell of Figure 4C represent P values, but in Figure 5C they represent correlation scores. Such inconsistencies in presentation may confuse readers. It might be helpful to adopt a more uniform design if possible.

12. Sulfur is an essential element in the human body, existing in various organic compounds (e.g., sulfur-containing amino acids, chondroitin sulfate, taurine). Unlike metallic nutrients such as sodium, potassium, or magnesium, sulfur is seldom discussed in terms of total quantity because of its diverse organic forms. Ideally—though it may go beyond the scope of your current study—estimating the total sulfur intake (absolute mass) or measuring sulfur levels (e.g., in serum, intestinal tissue, urine, or feces) via elemental analysis methods such as ICP-MS could provide further insight. For reference, methods for quantifying sulfur in biological samples by ICP-MS are discussed in detail in

https://doi.org/10.1039/C5JA00489F,

https://doi.org/10.1002/elps.202400128.

Furthermore, Okamoto et al. (J Cancer Epidemiol Prevent. 2020;5:1–9) also discuss relevant aspects of sulfur analysis, although the DOI is currently inactive (10.36648/cancer.5.1.4). If such approaches or similar published precedents exist, it would be worthwhile to review them for potential integration into future research on sulfur intake and ulcerative colitis management.

13. In the “Study Design and Participants” section, you reference Article 1 (Ref. 1) as containing details of the original trial design and the CONSORT diagram. Please verify that this citation is indeed correct. If not, kindly update it accordingly.

14. It appears that your team’s prior work is detailed in Reference 45. Ideally, the methodology should be fully reproducible without requiring readers to refer to that previous study. Additionally, you mention the registry number NCT04474561 in line 538; it may be beneficial to briefly explain this registry in the Introduction so readers can understand its significance early on.

Author Response

We thank the reviewer for their comments and suggestions which have improved the manuscript. We have resonded line by line to the reviewer's comments which is included in the Pdf.

Reviewer 2 Report

Comments and Suggestions for Authors

Review report of manuscript “Reduced Sulfur Diet Reshapes the Microbiome and Metabolome in Mild-Moderate Ulcerative Colitis

Dear authors

The idea of the manuscript is very good and introduced well. 

Results and conculsion are very important and inovated. As the authors mentioned in the manuscript the study contins many limitations. I think the most important limitations ar the sex of patients as most of patients are females and the short time of the intervention study. 

Some editing mistakes are found please correct it.

The introduction needs to be updated, the recent reference is in the year 2023. Please add more details about dietary sources of sulfur.

Delete year 2022 in L 486.

References need to be updated to the year 2025.

In the supplementary file, please add the units of Zonulin and Lipopolysaccharide Binding Protein in Table 4 S.

Author Response

Thank you to the reviewer for their comments and suggestions.  In the attached pdf we respond line by line to their comments.  Changes have been incorporated into the manuscript.

Round 2

Reviewer 1 Report

Comments and Suggestions for Authors

The authors have responded satisfactorily to the reviewers' comments and made sufficient corrections, making the paper worthy of acceptance.

Author Response

Thank you for your comments